# Wayfinding and path integration deficits detected using a virtual reality mobile app in patients with traumatic brain injury

Caroline Seton[1], Antoine Coutrot[2], Michael Hornberger[3], Hugo J. Spiers[4], Rebecca Knight[1]☯*, Caroline Whyatt[1]☯

1 Department of Psychology, Sport and Geography, University of Hertfordshire, Hatfield, Hertfordshire, United Kingdom, 2 Laboratoire d'InfoRmatique en Image et Systèmes d'information, French Centre National de la Recherche Scientifique, University of Lyon, Lyon, France, 3 Applied Dementia Research, Norwich Medical School, University of East Anglia, Norwich, United Kingdom, 4 Division of Psychology and Language Sciences, Department of Experimental Psychology, University College London, London, United Kingdom

☯ These authors contributed equally to this work.
* R.knight4@herts.ac.uk

**Data Availability Statement:** All relevant data are within the manuscript and its Supporting information files.

## Abstract

The ability to navigate is supported by a wide network of brain areas which are particularly vulnerable to disruption brain injury, including traumatic brain injury (TBI). Wayfinding and the ability to orient back to the direction you have recently come (path integration) may likely be impacted in daily life but have so far not been tested with patients with TBI. Here, we assessed spatial navigation in thirty–eight participants, fifteen of whom had a history of TBI, and twenty–three control participants. Self-estimated spatial navigation ability was assessed using the Santa Barbara Sense of Direction (SBSOD) scale. No significant difference between TBI patients and a control group was identified. Rather, results indicated that both participant groups demonstrated 'good' self–inferred spatial navigational ability on the SBSOD scale. Objective navigation ability was tested via the virtual mobile app test Sea Hero Quest (SHQ), which has been shown to predict real–world navigation difficulties and assesses (a) wayfinding across several environments and (b) path integration. Compared to a sub-sample of 13 control participants, a matched subsample of 10 TBI patients demonstrated generally poorer performance on all wayfinding environments tested. Further analysis revealed that TBI participants consistently spent a shorter duration viewing a map prior to navigating to goals. Patients showed mixed performance on the path integration task, with poor performance evident when proximal cues were absent. Our results provide preliminary evidence that TBI impacts both wayfinding and, to some extent, path integration. The findings suggest long–lasting clinical difficulties experienced in TBI patients affect both wayfinding and to some degree path integration ability.

**Funding:** The authors received no specific funding for this work.

**Competing interests:** The authors have declared that no competing interests exist.

## Introduction

The remarkably complex and dynamic system comprising several basic cognitive processes that subserve everyday navigational abilities is conceptualised as spatial cognition. Spatial cognition refers to the integration and interpretation of spatial information necessary for space-oriented behaviour—such as visual–guided prehension movement and attention orientation towards stimuli relevant to the construction of a mental representation of one's environment [1]. It is believed that wayfinding–a complex navigational behaviour that encompasses decision-making about navigational goals, as well as the formation and maintenance of intent to travel–is required to create a 'cognitive map' of how locations are related to each other in space [2]. The perception and processing of an environment's spatial characteristics–as facilitated by the visual, vestibular, and proprioceptive sensory systems–allows for the generation of this mental representation and is theorised to take place from either an egocentric or allocentric perspective [3, 4]. Allocentric representations pertain to the relationships between locations or landmarks that serve as points of reference irrespective of the observer's position in the environment. An allocentric mental representation, therefore, enables one to visualise the environment independent of one's viewpoint [5]. Egocentric representations consist of spatial information about the position of locations or landmarks in relation to the observer's position in the environment. Subsequently, learning to associate landmarks with a directional response that is dependent on one's viewpoint from a ground–level perspective is implicated in wayfinding from an egocentric perspective [6]. The dichotomy of attributing spatial navigation abilities to the allocentric and egocentric perspectives is, however, not comprehensive enough to conceptualise all the mechanisms needed in spatial navigation. Indeed, allocentric and egocentric navigation mechanisms seamlessly interact during real–world navigation. However, while artificial, the splitting of, and thus independent assessment of, each is often necessary for experimental studies.

Given the importance of navigation in everyday functioning, the neurocognitive mechanisms upon which this ability is dependent has received significant interest [7]. It is well-established that spatial cognition is dependent on the integrity of the brain structures that underpin the complex interactions between multiple cognitive processes that are required for successful navigation. As with most cognitive tasks, these include attention, working memory, spatial processing, planning, decision-making, and error monitoring [8]. Such cognitive complexity renders spatial navigation particularly vulnerable to brain injury [9]. Traumatic brain injury (TBI) refers to the focal or diffuse disruption of the functional and structural integrity of the brain caused by an external physical force transmitted to the head [10]. Topographical disorientation is a frequently reported complaint of TBI patients, with moderate to severe injury often leading to a generalised impairment in spatial cognition in both familiar and unfamiliar environments [11]. Moreover, object-location memory may be particularly vulnerable to the lasting effects of TBI, further exaggerating spatial navigation difficulties [9].

Over the last two decades, virtual reality (VR) has been a critical tool in empirical investigations into the structural and functional neuroanatomical correlates of spatial navigation abilities due to the accurate replicability of the virtual environment's properties [12]. Numerous group studies have demonstrated a good concordance across real-world and VR environments in spatial navigation performance in a healthy population [13], in younger and older age groups [14], in individuals with brain injury [15], and chronic stroke patients [11]. Furthermore, VR tests of spatial cognition have been shown to be more sensitive in identifying spatial navigation deficits in clinical patient populations compared to more classic visuospatial 'pencil–and–paper' tests [16, 17].

Tu and colleagues explain that the advantages of evaluating navigation abilities in virtual environments as opposed to real–world situations include that it allows for the measurement of navigation performance that would otherwise be impeded by physical disability [18]. This advantage can be considered at two levels. First, the application of VR environments presented on tablets and smartphones facilitate a unique approach to studying and quantifying spatial navigation abilities, as it allows for remote access to a sample of participants that would otherwise be limited by geographical location. Second, the utilisation of VR technology allows for the measurement of ecological navigation behaviour and real–time interactions within complex virtual environments, including for individuals with physical limitations [19]. Spatial navigation metrics that are typically drawn from VR environments include: accuracy in navigating the virtual environment, time taken to complete the navigation task (also referred to as latency), and distance travelled within the virtual environment [20]. Distance and duration metrics on wayfinding tasks have been used in clinical settings to measure error whereby, these metrics efficiently characterise participant's profiles: slow/fast and short/long distance travelled [20].

Despite the versatility of VR in studying spatial cognition, a review of the literature pertaining to the use of VR for the assessment of spatial navigation deficits in TBI patients yields limited findings. Available empirical evidence pertaining to wayfinding deficits after TBI does indicate that several cognitive processes may be impaired, such as the ability to maintain the intention to reach a particular location and utilising egocentric navigation strategies to guide wayfinding [21]. More specifically, TBI patients have been found to demonstrate an impairment in the ability to learn and remember the configurations of an environment from an allocentric perspective to form a cognitive map, which has been supported by the small number of published empirical studies [2, 22–24]. However, results infer two potential moderating factors in the presentation of spatial difficulties. First, individuals with mild TBI demonstrate no significant difficulties in spatial navigation despite presenting clinical symptoms such as loss of concentration, dizziness, fatigue, headache, irritability, and visual disturbances [25]. This raises questions over the intuitive impact of brain injury severity, but also the impact of clinical symptoms which may directly impact upon performance by moderating VR engagement. A recent study found that forced choice decisions regarding landmark location and route direction were impaired in patients with acquired brain injury [26]. The study used a desktop virtual reality paradigm where patients viewed a series of first-person perspective wayfinding videos. This large-scale online study was conducted on 435 patients, whose mechanism of injury included cerebrovascular accidents, TBI, brain tumours, infection, intoxication, epilepsy, multiple sclerosis, and hypoxia. When drawing conclusions, it is important to consider the variable nature of deficits associated with these pathologies and the impact thereof on navigation ability. As noted by Banville and colleagues, manipulating a device for VR interaction and understanding the rules that are necessary to correctly navigate within the virtual environment places an additional demand on cognitive processes such as memory, attention and planning [20]. There is, however, no evidence pertaining to the impact these lasting clinical difficulties may have on spatial navigation abilities in individuals who have suffered a TBI. Second, the type of spatial navigation task may modulate the consistency of findings. Recent studies demonstrated the transferability of navigation performance in real–life and virtual environments for landmark recognition and route distance estimate tasks, but not in tasks such as drawing a map of a virtual navigation route or path integration [12, 27].

This present study was exploratory and conducted in collaboration with a charity in order to reach out to patients. We used the VR navigation test Sea Hero Quest (SHQ; [28, 29]) to measure spatial navigation ability of participants with TBI. SHQ is an app for mobile

devices involving navigating a boat through aquatic environments. It features two main tasks, wayfinding and path integration, which have been designed to measure different aspects of spatial navigation [28]. The use of the SHQ is of particular benefit by which participant data from the current study can be compared to the population benchmarks obtained in a previous study using SHQ, in which the navigation performance of 3.9 million people worldwide was assessed [12, 28]. SHQ has been shown to accurately predict real-world navigation performance [12]. In addition, we also examined subjective self-estimates of navigation ability using the fifteen–item Santa Barbara Sense of Direction (SBSOD; [30]).

To summarise, the results of the present study reveal that the group differences on the wayfinding tasks which provide evidence suggesting TBI participants show a pronounced and consistent wayfinding impairment are further strengthened by contextualising the findings and showing that the control group median is in-line with the more generalisable population results. The findings of the present study are also in support of the evidence which states that VR tests of spatial cognition have been shown to be more sensitive in identifying spatial navigation deficits in clinical patient populations compared to self-reported measures.

## Materials and methods

### Participants

Opportunistic sampling was utilised for the recruitment of participants whereby an invitation to participate in the study was advertised by the "Brain Injury Support (BIS) Services" which is a private organisation which provides specialist cognitive rehabilitation therapy to individuals with brain injury, and "High Beyond C" which is an organisation that facilitates an interactive virtual program for brain injury survivors.

Thirty–eight (n = 38) participants were recruited to the study—fifteen (n = 15) with a history of TBI (mean age = 31.67 ± 12.34 yrs.) and twenty–three participants (n = 23) from the general population to serve as healthy controls (mean age = 32.61 ± 12.59 yrs.).

From the sample of thirty–eight participants that completed the SBSOD scale, a subsample of ten participants from the TBI group (n = 10) and a subsample of thirteen participants from the control group (n = 13) completed the SHQ navigation tasks. Thus, while the participant group remains small, this number is inline previous empirical studies; 14 TBI and 12 controls [23], TBI and 12 control [2], and eight TBI and 40 control [24]. See Table 1 for the demographic characteristics of the participants in the overall sample, as well as the subsamples that completed the SHQ game.

The research conducted in this study was undertaken in concordance with the University of Hertfordshire Health, Science, Engineering and Technology Ethics Committee with Delegated Authority. The ethics protocol number for this study was LMS/PGT/UH/04139.

Participants with a history of TBI (n = 15) disclosed the year in which they acquired the brain injury and the type of TBI; nine participants acquired a closed head injury, one

**Table 1. Demographic characteristics of TBI and control participants.**

| | | Male | | Female | | Age | |
|---|---|---|---|---|---|---|---|
| | | n | % | n | % | Mean | SD |
| Total Sample | TBI | 8 | 53.3% | 7 | 46.7% | 31.67 | 12.34 |
| | Control | 6 | 26.1% | 17 | 73.9% | 32.61 | 12.59 |
| SHQ Subsample | TBI | 5 | 50% | 5 | 50% | 28.31 | 5.92 |
| | Control | 5 | 38.5% | 8 | 61.5% | 27.6 | 10.46 |

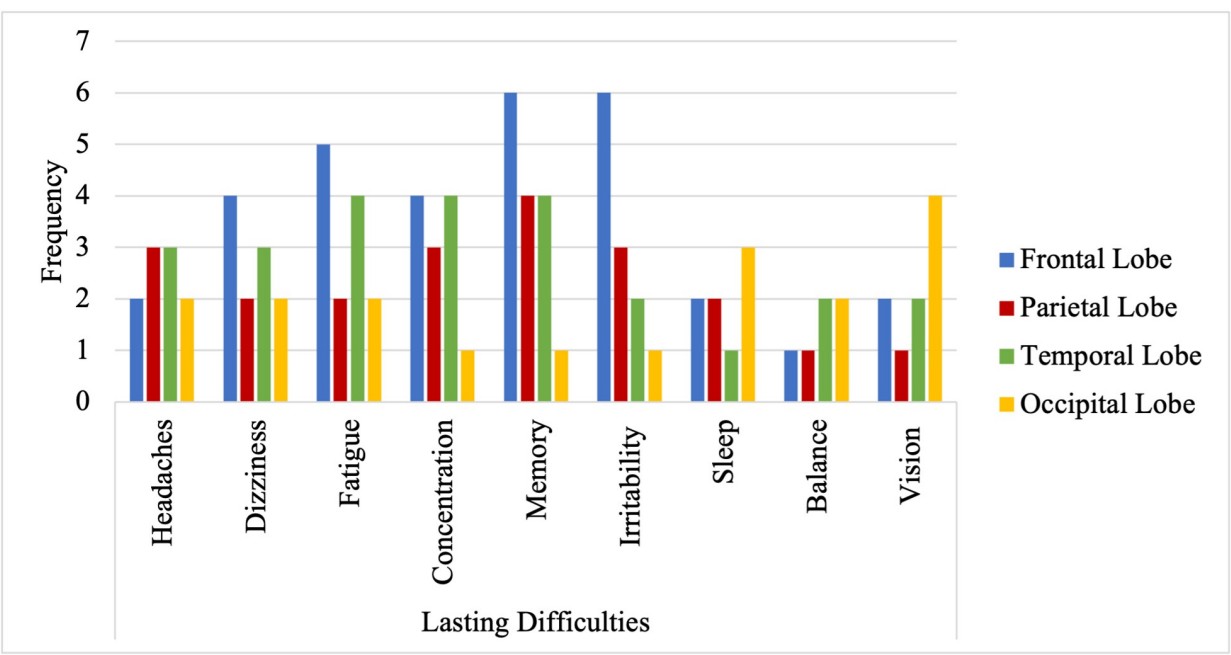

**Fig 1. Frequency of self–reported lasting difficulties according to self-reported location of damage.**

participant acquired an open head injury, two participants acquired a skull fracture and lastly, three participants reported acquiring the TBI as a result of an 'other' mechanism of injury. Participants with a history of TBI also provided a self-report disclosing the location to which the injury was acquired (i.e., frontal lobe, temporal lobe, parietal lobe, or occipital lobe) and whether they experience persistent difficulties due to the TBI (i.e., headaches, dizziness, excessive physical or cognitive fatigue, concentration, memory, irritability, sleep, balance, vision or other). See Fig 1 for a summary of the reported frequencies of lasting difficulties according to self-reported location of damage.

## Design

This study compared navigation ability between two groups (TBI or control) using three spatial navigation measures, namely the SHQ wayfinding duration and distance, SHQ path integration, and SBSOD scale. The first and second dependent variables, specific to performance on the wayfinding tasks, were duration (as measured by time taken in seconds to complete each level), and distance travelled (quantified as the Euclidean distance travelled between each sampled location in each level (in pixels)). Note, the coordinates of participants' trajectories were sampled at Fs = 2 Hz. In line with the characterisation of wayfinding performance [12] better performance is represented by lower duration and distance metrics. The third dependent variable, specific to performance on the path integration tasks, was flare accuracy (i.e., selection of correct starting location in path integration tasks from three options). Lastly, the fourth dependent variable was participants' scores on the fifteen–item SBSOD questionnaire, which range from 1 to 7, with higher scores indicating better self–rated navigational abilities [30].

## Materials

**Sea Hero Quest.** The Sea Hero Quest (SHQ) application features two main tasks, wayfinding, and path integration [28, 29]. The wayfinding tasks presented via the SHQ application

start with an allocentric view of a map that displays the participant's current starting location and the locations of the goals to find once the game commences. There were no time restrictions for the duration that participants may study the map. The wayfinding task entails navigating a boat along a river to the goal locations in the order indicated by the map (e.g., goal one must be found first, then goal two, and so on). Goals are identifiable as buoys with flags marking the goal number. The task is complete when all goals have been located. These tasks have been shown to require cognitive processes such the interpretation of a map, planning a multi–stop route, learning and remembering the navigation route, monitoring progression along the route, as well as the transformation from an allocentric to an egocentric perspective needed for navigation [12, 31]. The real–world ecological validity of the SHQ way finding tasks has been demonstrated by a significant correlation (r = 0.44) between navigation performance on the virtual wayfinding tasks and real–world city street finding tasks [12]. More specifically, Coutrot and colleagues [12] report a strong correlation between the distance participants travelled in the video game (in pixels) and in the real–world street network (in metres, measured by a GPS device). Wayfinding performance metrics derived from SHQ include duration as measured by time taken and distance quantified as the Euclidean distance travelled within the goals. The first two levels completed by a participant were treated as training levels, as they were only designed to assess ability to control the boat (i.e., tap left to turn left, tap right to turn right, swipe up to speed up and swipe down to halt). A subset of five levels that varied in difficulty were selected to evaluate wayfinding performance, each with increasing levels of difficulty.

Performance on the virtual wayfinding and path integration tasks are argued to be dependent on different cognitive processes, and this therefore accounts for the previous finding that performance on the virtual path integration tasks (as measured by flare accuracy) was not significantly correlated with performance in a real–world street network [12]. Specifically, the path integration tasks are thought to rely upon the perception of ego–motion during navigation, which serves to update one's orientation to the virtual environment and is typically dependent on spatial and working memory [32]. The path integration tasks presented via the SHQ app entails navigating along a river with bends until the participant identifies a flare gun, at which point the boat rotates by 180˚. The participant then chooses to shoot the flare in one of three directions (right, front, and left) that they believe points to the correct starting location. They are awarded stars for this choice, three stars for the correct answer (correctly selecting the direction referencing the starting point), two stars for the second closest direction, and one star for the third closest direction. A subset of three levels that varied in route complexity and thus difficulty was selected to evaluate path integration performance, with each level increasing in the number of transverse bends (one, three and four). See Fig 2 for visual examples of wayfinding and path integration tasks presented by the SHQ application.

**Santa Barbara Sense of Direction Scale.** The Santa Barbara Sense of Direction Scale (SBSOD; [30]) is a fifteen–item questionnaire based on a seven–point Likert scale where responses range from *strongly agree* to *strongly disagree* was used to assess spatial and navigational abilities, preferences, and experiences such as giving directions and reading maps. Scores on the SBSOD scale range from 1 to 7, with higher scores indicating better self–rated navigational abilities [30]. The SBSOD scale has demonstrated adequate reliability (Cronbach's $\alpha$ = 0.88) and high test–retest reliability (r = 0.91). Collectively, results from validity studies indicate that the SBSOD scale is strongly related to objective measures of spatial navigation skills, whereby SBSOD scores have been shown to correlate with accuracy in identifying landmark locations within a localised environment, learning the layout of a new environment through actual experience, and with learning and navigating within a new VR environment [33]. The items of the SBSOD questionnaire measure the following domains of spatial and

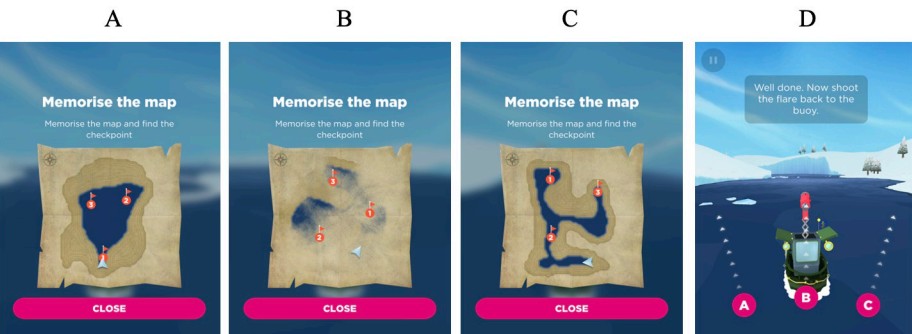

**Fig 2. Sea Hero Quest visual examples of wayfinding and path integration levels.** Wayfinding levels 6, 8 and 11 presented in A, B, and C respectively. Path integration level 4 is presented in D.

navigational abilities: giving directions to others (items one and 11); receiving and using directions (item eight); memory for routes (items 10 and 14); using and liking maps, GPS, or other navigation technology, in the case of 'planning' a route (items seven, nine, and 13). The remaining items cover other specific aspects of spatial awareness: distance (item three), actual 'sense of direction' as interpreted by the respondent (item four), cardinal directions (item five), novel environments (item six), location awareness (item 12) and finally, maintenance of a 'mental map' (item 15) [34]. All responses were scored according to the scoring rubric, with the reverse scoring of positively worded items.

## Procedure

The study was conducted remotely, whereby participants received an invitation to participate in the study, which directed the participants to an online Qualtrics survey. First, participants were asked to read and complete a standard informed consent form, followed by prompts for participants to enter demographic information such as date of birth, gender and whether they have a history of a TBI. Participants that answered that they had a history of a TBI were prompted to enter the year of TBI event and were presented with additional multiple–choice questions pertaining to 1) the type of TBI: closed head injury, open head injury, skull fracture or other; 2) the location to which the injury was acquired: frontal, temporal, parietal, occipital or other; 3) if they had persistent difficulties as a result of the TBI: headaches, dizziness, excessive physical or cognitive fatigue, concentration, memory, irritability, sleep, balance and vision. The Qualtrics survey then presented the items of the SBSOD questionnaire. Upon completion of the questionnaire, participants were prompted to download an instruction sheet which outlined the step–by–step process to download the SHQ game, create a session ID and the ten levels of the game the participants were asked to play. Upon completion of playing the SHQ levels, participants were guided by the instruction sheet to export the game data to an email account created specifically for the purposes of this study. A code was developed allowing for the SHQ performance data to be extracted using MATLAB (R2020a) and analysed using SPSS (Version 26).

## Results

### Santa Barbara Sense of Direction Scale

Given the evidence suggesting the validity of the SBSOD scale as a measure of self-reported wayfinding ability, it is of interest to determine whether there is a relationship between

perceived wayfinding ability and empirically assessed wayfinding ability as measured using SHQ. The sample of participants with a history of TBI (n = 15) obtained a mean SBSOD score of 4.00 (SD = .93, SE = .24) and the control sample of participants without a history of TBI (n = 23) and obtained a mean SBSOD score of 4.35 (SD = 1.07, SE = .22). Data were found to violate assumptions for parametric analysis, and thus the difference between groups was ana-lysed using Mann–Whitney U Test, which showed that the TBI and control groups' SBSOD scores were not significantly different (U = 140.50, $p$ = .34), with median scores of 4.00 for both groups. The SBSOD scores of the TBI and control groups therefore did not significantly differ; rather, both participant groups demonstrated 'good' self–inferred spatial navigational ability on the SBSOD scale as characterised by higher scores [30].

## Sea Hero Quest

**Wayfinding.** To examine whether there were differences between groups' general ability to use the SHQ app and game interface, an independent samples t-test was used to determine if there were differences in baseline wayfinding performance, as measured mean duration scores on the first two training levels—designed only to assess ability to control the boat—of the SHQ application. There was not a significant difference ($t$(21) = —.52, $p$ = .61) on the mean wayfinding performance between the participants with a history of TBI (n = 10) and control participants (n = 13), with mean scores of 23.65 and 22.71 respectively. As these two training levels were not designed to assess wayfinding ability, this finding infers that both groups are interacting with the game controls in a similar way, thus subsequent differences in wayfinding levels are unlikely to be due to participants having difficulty using the interface.

To determine the overarching group differences between the TBI and control groups per-formance on SHQ's respective wayfinding tasks, overall distance and duration metrics were calculated. The mean overall distance (quantified, in pixels, as the Euclidean distance travelled between each sampled location in each level at Fs = 2 Hz) on all wayfinding levels for control participants (n = 13) was 736.50 (SD = 100.52, SE = 27.91). The mean overall distance on all wayfinding levels for participants with TBI (n = 10) was 1010.77 (SD = 211.36, SE = 66.84). The results of an independent samples t–test demonstrated that the difference between the TBI and control groups' mean overall distance on the wayfinding levels (*-274.28*, 95% CI: *-412.39* to *-136.16*) was significant $t$(21) = -4.13, $p$ < .001, 2-tailed. Equal variances were assumed and represented a large effect size (Cohen's $d$ = -1.74). The mean overall duration on all wayfinding levels for participants with TBI was 78.48 seconds (SD = 25.34, SE = 8.01). The mean overall duration on all wayfinding levels control participants was 53.70 seconds (SD = 15.95, SE = 4.42). The results of an independent samples t–test demonstrated that the difference between the TBI and control groups' mean overall duration on wayfinding levels (*-24.69*, 95% CI: *-42.62* to *-6.75*) was significant $t$(21) = -2.86, $p$ = .005, 2-tailed. Equal variances were assumed and represented a large effect size (Cohen's $d$ = -1.20). These findings infer that the TBI participants had a more pronounced wayfinding impairment overall.

To explore this further, a nuanced approach was taken, exploring performance on a level-by-level analysis. To this end, Mann-Whitney U test was employed to determine if there were differences in wayfinding performance as measured via the individual levels of the SHQ appli-cation between participants with a history of TBI (n = 10) and control participants (n = 13). Table 2 demonstrates that the distance metrics on all levels of difficulty were significantly dif-ferent between the TBI and control groups, with the TBI group taking a significantly longer route.

Further, while overall results indicate a clear trend for individuals with TBI to consistently take a longer time to complete each level, again indicative of poorer wayfinding performance,

**Table 2. Results of Mann–Whitney U test on distance and duration metrics on all wayfinding tasks between TBI and control group.**

| Wayfinding Levels | TBI | Control | U | z | p value |
|---|---|---|---|---|---|
| | (Median) | (Median) | | | |
| **Distance** | | | | | |
| Level WF_6 | 332.84 | 302.04 | 110.00 | 2.79 | .01* |
| Level WF_8 | 975.46 | 632.49 | 102.00 | 2.30 | .02* |
| Level WF_11 | 1138.67 | 790.17 | 104.00 | 2.42 | .02* |
| Level WF_16 | 944.53 | 757.18 | 103.00 | 2.36 | .02* |
| Level WF_46 | 1642.64 | 1045.49 | 109.00 | 2.73 | .01* |
| **Duration** | | | | | |
| Level WF_6 | 36.75 | 24.50 | 91.50 | 1.64 | .10 |
| Level WF_8 | 73.25 | 41.00 | 102.00 | 2.30 | .02* |
| Level WF_11 | 85.50 | 61.00 | 95.50 | 1.89 | .06 |
| Level WF_16 | 61.50 | 51.50 | 80.50 | .96 | .34 |
| Level WF_46 | 136.00 | 67.50 | 111.00 | 2.85 | .004* |

Distance measured by Euclidean distance travelled in pixels and duration measured in seconds.

* $p < 0.05$.

these differences only reach significance for only two out of the five completed levels. To this end, the time spent viewing the map–a factor which may relate to how accurately the map is encoded and later recalled–was explored to assess the relative importance of this strategic activity. Participants with a history of TBI (n = 10) had a median overall map view duration of 4.12 seconds and control participants (n = 13) had a median overall map view duration of 9.89 seconds. The results of a Mann-Whitney U test demonstrated that there was a significant difference in median overall map view duration (U = 32, $z$ = - 2.05, p = .04). The results of further exploratory analysis for each of the individual levels of the SHQ application suggest that TBI participants consistently viewed the map for a shorter duration prior to commencing the wayfinding task, although these differences only reach significance for only two out of the five completed levels (see Table 3).

**Comparison to population data.** To contextualise these findings, Fig 3 indicates that the control group median (indicated by the dark line) is in-line with the larger on-line population (ages 20 to 30 years) results upon which the wayfinding metrics are plotted against (N = 15275). Contrastingly, the TBI group (individually plotted by dashed lines with the exception of a participant exceeding the above age range) shows greater dispersion across the population metrics, and notably away from the population skewed mean, and control group median. Upon visual inspection of Fig 3, TBI participants' performance on the training levels upon which baseline performance was measured, is in accordance with that of the control

**Table 3. Results of Mann–Whitney U test on map view duration on all wayfinding tasks between TBI and control group.**

| Wayfinding Levels | TBI | Control | U | z | p value |
|---|---|---|---|---|---|
| | (Median) | (Median) | | | |
| Level WF_6 | 1.65 | 5.00 | 30.00 | -2.17 | .03* |
| Level WF_8 | 4.19 | 9.04 | 39.00 | -1.61 | .11 |
| Level WF_11 | 2.60 | 12.80 | 30.50 | -2.14 | .03* |
| Level WF_16 | 5.97 | 8.09 | 41.00 | -1.49 | .14 |
| Level WF_46 | 6.15 | 16.12 | 37.00 | -1.74 | .08 |

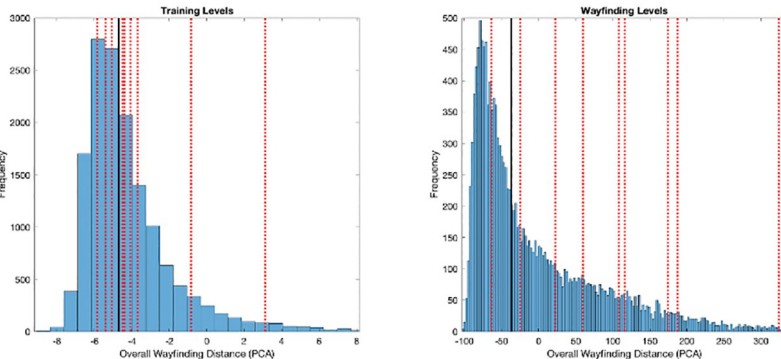

**Fig 3. Summary histogram of control group and TBI group distance performance on training and wayfinding levels plotted against population data.**

group, however variability is clearly identifiable across wayfinding levels which is in accordance with the results discussed earlier.

More specifically, Fig 4 pertains to a histogram in which the performance metrics TBI participant (patient A, a 21 y.o., Male), who consistently demonstrated the poorest wayfinding ability, as represented by overall distance score, are plotted against the wayfinding metrics of a

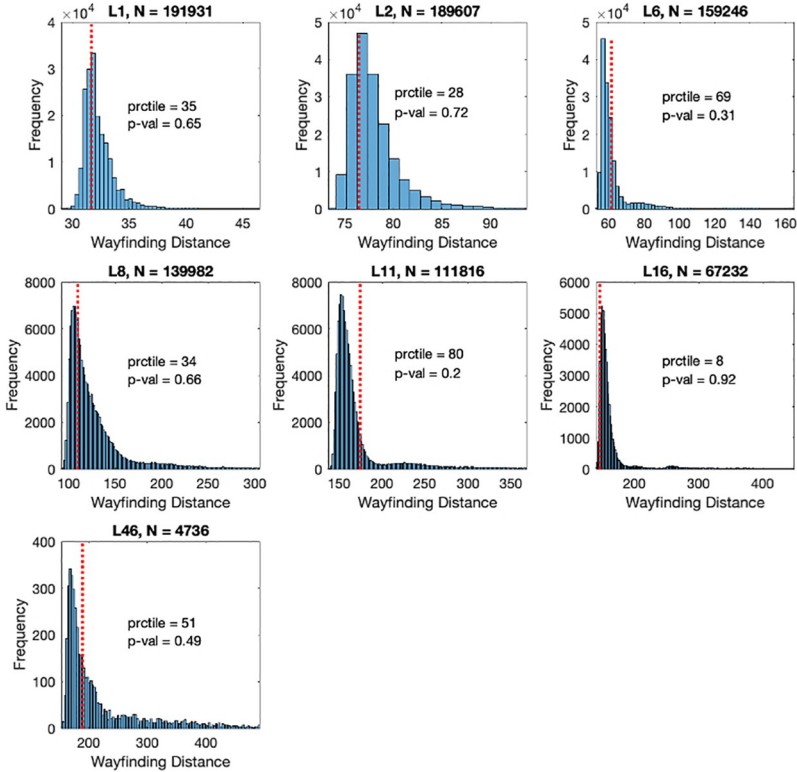

**Fig 4. Histogram of distance metrics of patient A individually plotted against distance metrics of control population.** 'N' is the control sample size. The red dashed vertical line corresponds to the patient's performance. Prctile is the percentile of the patient's performance in the control distribution. P-value is the inverse (number of controls with worst performance / N).

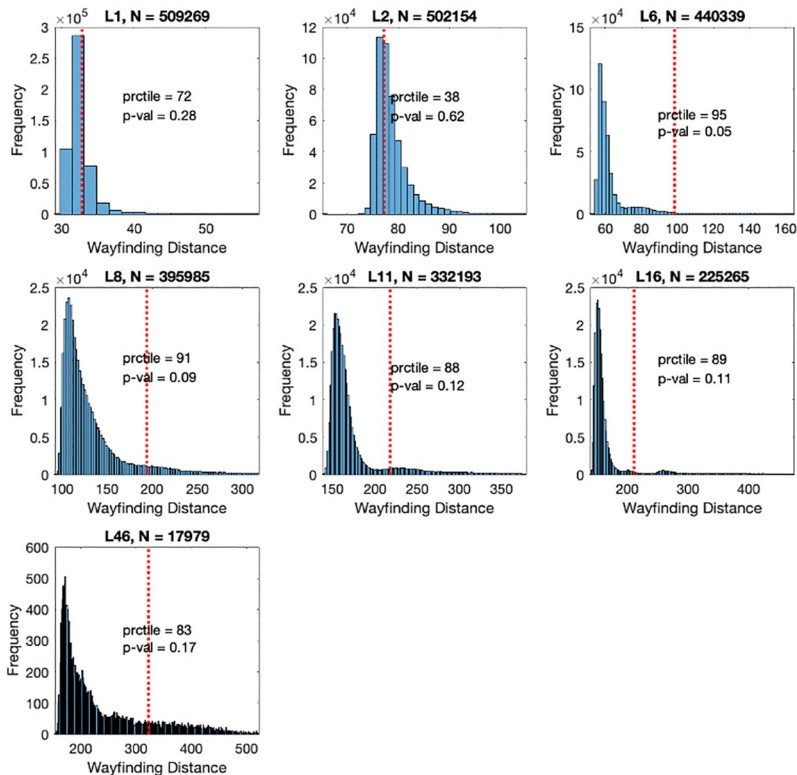

**Fig 5. Histogram of distance metrics of patient B individually plotted against distance metrics of control population.** 'N' is the control sample size. The red dashed vertical line corresponds to the patient's performance. Prctile is the percentile of the patient's performance in the control distribution. P-value is the inverse (number of controls with worst performance / N).

population of control participants with the same age (± 3 years) and gender. Participant A also reported experiencing lasting difficulties following TBI and includes physical and cognitive fatigue, concentration difficulties, memory impairment and sleep disturbances.

Contrastingly, Fig 5 pertains to a histogram in which the performance metrics TBI participant (patient B, a 25 y.o., Male), who demonstrated wayfinding ability that is consistently more within 'normal' range are plotted against the wayfinding metrics of a population of control participants with the same age (± 3 years) and gender. Patient B also reported experiencing lasting difficulties following TBI, including dizziness and irritability. Notably, the association between increased number of lasting difficulties and performance metrics on the wayfinding tasks was not significant, nor were there specific lasting complaints associated with poor wayfinding.

**Path integration.** Variation in path integration ability of the TBI participants was uncovered when examined using the VR SHQ task. Specifically, participants that completed both the SBSOD scale and path integration levels consisted of a group of participants with a history of TBI (n = 10) and a group of participants with no history of TBI (n = 13). To determine the likelihood of participants' selecting the correct starting location during the path integration tasks (as a measure of accuracy), the following ratios were found. As can be seen in Fig 6, one participant (10%) did not select the correct starting location on any of the three path integration levels. Three TBI participants (30%) and one control participant (7.7%) selected the correct starting location once. Six TBI participants (60%) and eight control participants (61.5%)

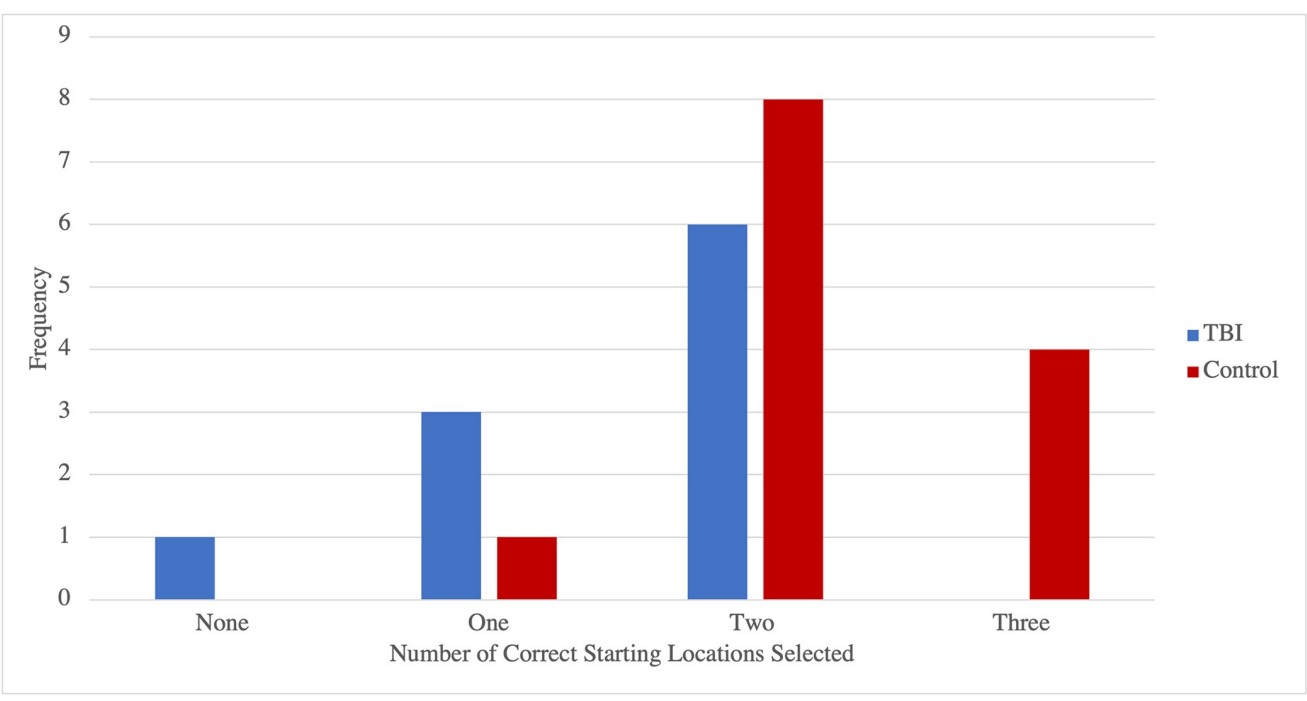

**Fig 6. Frequencies of TBI and control participants as a function of overall flare accuracy on path integration levels.**

selected the correct starting location twice. Four control participants (30.8%), but no TBI participants, selected the correct starting location on all three path integration levels. The results of a Fisher exact test showed a significant, moderate association between path integration accuracy and history of TBI, $p = .05$ (2-tailed), Cramer's V = .46. These findings suggest that participants without a history of TBI are more likely to demonstrate an ability to perform path integration tasks with a higher degree of accuracy.

On level PI_4, 60% of the TBI group selected the correct starting location and 92.3% of the control group selected the correct starting location. As such, control participants were eight times more likely to achieve the correct flare accuracy than participants with a history of TBI (OR = 8.00), however the results of a Fisher exact test demonstrated that the association between flare accuracy and history of TBI on Level PI_4 was not significant, $p = .13$ (2–tailed).

Similarly, on Level PI_34, 40% of the TBI group selected the correct starting location and 84.6% of the control group selected the correct starting location. Thus, control participants were eight times more likely to achieve the correct flare accuracy than participants with a history of TBI (OR = 8.25). The results of a Fisher exact test showed a significant association between flare accuracy on Level P1_34 and history of TBI, $p = .03$ (2–tailed). There was a moderate association between flare accuracy and history of TBI, Cramer's V = .46.

Despite the above, on the most difficult path integration level (Level PI_54 with four bends) 30% of the control group selected the correct starting location and 46.2% of the TBI group selected the correct starting location, inverting expectations. Participants with a history of TBI were twice as likely to achieve the correct flare accuracy than the control participants (OR = 2). Perhaps unsurprisingly, the results of a Fisher exact test demonstrated that the association between flare accuracy on Level PI_54 and history of TBI was not significant, $p = .67$ (2–tailed). It may be that this final level was simply too challenging for any participant.

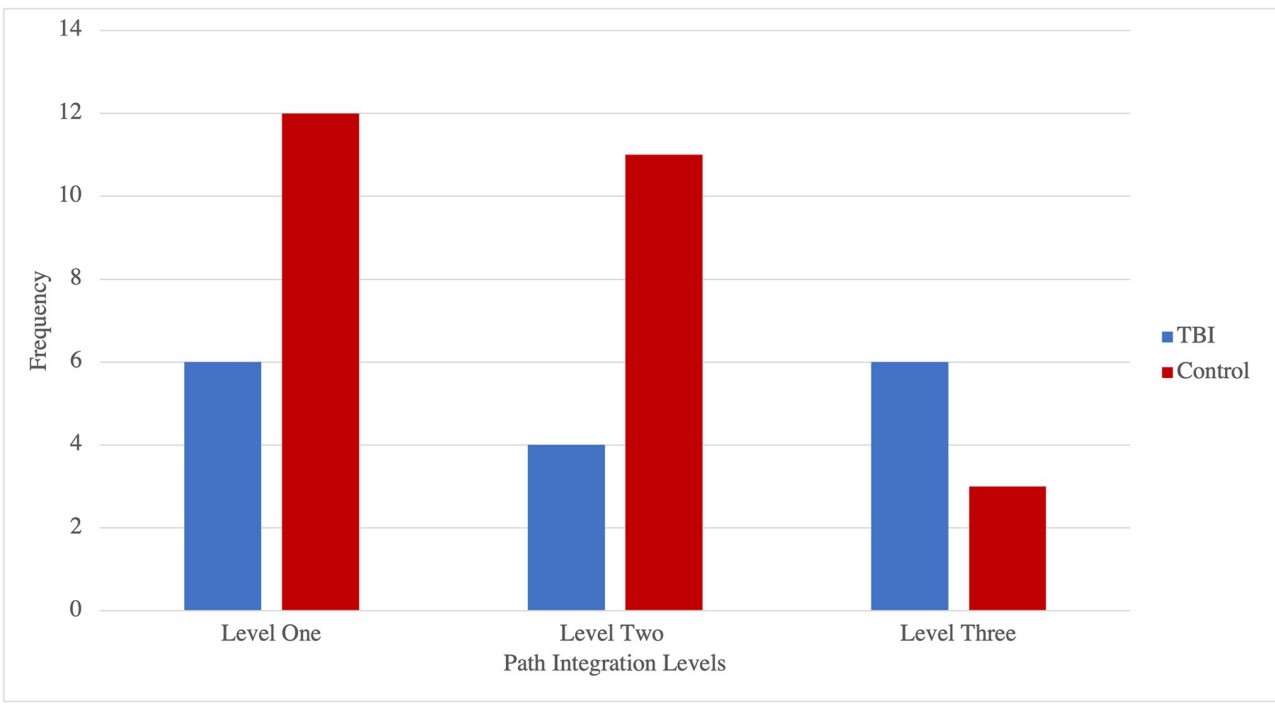

**Fig 7. Frequencies of TBI and control participants as a function of flare accuracy (i.e., selection of correct starting location) on path integration levels.**

Fig 7 demonstrates the frequencies of TBI and control participants as a function of flare accuracy (i.e., selection of correct starting location in path integration tasks).

## Discussion

Previous literature suggests that individuals who have sustained a TBI may present with an all–encompassing impairment in spatial navigation [2, 15]. Therefore, the hypothesis that there would be a significant difference in performance metrics (i.e., distance and duration) on way-finding tasks between the TBI and control groups is supported by the results in which TBI participants were found to show a more pronounced wayfinding impairment on overall performance metrics. A more nuanced approach was taken by comparing performance on a level-by-level analysis and indicated a significant difference in distance metrics on all levels of the wayfinding tasks between the TBI and control participant groups, in which the TBI group consistently took significantly longer routes. The findings that TBI participants travelled significantly longer distances within all the wayfinding levels than the control participants may provide support for the findings Carelli et al. who reported that TBI patients show an impairment in transferring and utilising knowledge about the configuration of the virtual environment in actual navigation performance within the virtual task [24]. Performance on the wayfinding tasks is argued to be dependent on various cognitive processes, which includes the ability to remember the navigation route. Further evidence supporting this conclusion is the finding that TBI participants consistently viewed the wayfinding map prior to the task for shorter durations, which may have impeded accurate encoding of the route. Although group differences in map view durations were significant overall, a level-by-level analysis indicates that the impact of these shorter encoding durations on wayfinding ability may be complex.

Within this study, group differences in map viewing duration reached significance on levels 6 and 11 whereas group differences on wayfinding performance were significant for levels 8 and 46.

While overall results indicate a clear trend for individuals with TBI to consistently took a longer time to complete each wayfinding level, which is indicative of poorer wayfinding performance, as mentioned above these differences only reach significance for two out of the five completed levels. These isolated significant differences in duration may be accounted for by the fact that the wayfinding tasks are becoming increasingly more challenging in terms of the complexity of the route travelled to reach the location of the target goals. Consideration of the requirements in different levels helps shed some light on the potential reasons for differences in performance. In level WF_8 to reach the locations of the goals, one is required to turn back towards the starting point to navigate towards the second goal. Furthermore, this level is the first of the wayfinding tasks where participants are required to navigate along a route with multiple bends, as Level WF_6 constitutes navigating within what can be described as an expanse of water without any bends. Similarly, Level WF_46 is arguably the most challenging of the wayfinding tasks, including an expanse of water with the target goals located within three of five channels through which participants are required to navigate. These considerations may account for the finding that the TBI participants took a longer time than the control participants to successfully complete these wayfinding levels. Future research will be able to explore this.

The results of this study may further postulate that the significantly greater distance metrics in the TBI participants are indicative of an impairment in object–location memory. The TBI participants' poorer performance on the wayfinding tasks may be explained by a difficulty with processing and integrating object–identity information (*what*) and object–position (*where*) from the map viewing, needed to construct a plan for navigation [9]. Such problems could plausibly lead to the errors in navigation. Beyond this initial encoding difficulty, it is possible TBI cases may have difficulty with a range of 'on-the-fly' processing required to actively navigate, such as mentally tracking travel compared to a plan, spotting the correct goal locations, avoiding recently encountered dead-ends etc. Our findings also agree with the results of van der Kuil and colleagues who found that individuals with ABI show impairments in landmark recognition and allocentric location knowledge and route-based path knowledge [26]. Taken together, there seems to be a consistent pattern of deficits in wayfinding / route finding in TBI patients for both active "game play" navigation (in this current study) and passive "video watching" navigation [26].

The study aimed to explore whether there were differences in performance metrics on the path integration task between the participants with a history of TBI and the control participants. Accurate performance on the path integration tasks is argued to be dependent on the perception of ego–motion during navigation, which serves to maintain and update one's orientation to the spatial characteristics of an environment [12]. It is argued that TBI patients may present with impairments in spatial and working memory, which are believed to be two of the underlying cognitive processes upon which spatial orientation is dependent [32]. We found a mixed pattern of performance on the path integration task. More specifically, results indicated a significant association between accuracy performance and having a history of TBI, but on a single level only (Level PI_34). Considering the properties of the levels, we note that level PI_34 has a more open visa terrain than the other levels which occur with local walls and would provide a stronger input of visual self-motion. Thus, while speculative, it is possible that TBI participants found PI_34 more difficult due to the weaker self-motion cues. Ultimately, more research will be needed to explore this possibility. These results suggest that the problems with navigation caused by TBI may go beyond the challenge of memorising a map and

recalling that during active wayfinding, but may extend to more egocentric processes, such as path integration, consistent with suggestions of a broader deficit in spatial processing in TBI [21].

As the Santa Barbara Sense of Direction (SBSOD) scale was developed as a reliable and valid measure of self-reported spatial navigation ability [34], we aimed to determine whether there were differences in mean SBSOD scores between participants with a history of TBI and control participants. Despite previous evidence demonstrating that individuals with a history of TBI scored significantly worse on self-report navigation and orientation, and distance estimation measures [26], no significant difference between TBI patients and a control group on the SBSOD was identified. Rather, results indicated that both participant groups demonstrated 'good' self–inferred spatial navigational ability on the SBSOD scale. Davies and colleagues argue that higher scores on the SBSOD scale are representative of one's self–confidence in utilising route–based spatial navigation (i.e., the ability to navigate within an environment from an egocentric perspective), rather than strongly related to the allocentric navigation process [34]. Furthermore, de Rooij et al. [35] argue that as the SBSOD negates the measurement of spatial anxiety which negatively impacts navigation ability in clinical populations. The limitation of self-reported instruments ought to be highlighted, as the scores rely on the accuracy of the patients' insights. Patients with brain injury are evidenced in many cases to have diminished insight into their actual cognitive and navigation performance in daily life [36]. Our findings agree with this perspective.

The findings of the present study support the view that VR tests of spatial navigation are more sensitive in identifying spatial navigation deficits in clinical patient populations than self-report measures. A similar argument is made for more classic visuospatial 'pencil–and–paper' tests, where-by these do not capture the navigation problems [16, 17].

## Limitations

A noteworthy limitation of the present study includes the utilisation of self–reports to collect data pertaining to the nature of participants' TBI, such as the location of damage. This limitation does make it difficult to assess the homogeneity of the sample. Despite the lack of objective measures, this study has allowed us to gain insight into spatial navigation impairment in TBI using a low cost, remote paradigm. In addition, the self-reported location of damage was not a significant predictor of the performance metrics in this study. Future studies could make use of neuroimaging and clinical records to support the classification of participants' injuries. Similarly, this limitation is relevant to the use of self-reported sense of direction ratings in the evaluation of participants' spatial and navigational abilities. Individuals with a history of TBI have been reported to demonstrate poor self–awareness [37], thus may have limited insight into the true nature of their navigational abilities. Future research into the nature of spatial cognition deficits in individuals with TBI could therefore involve the comparative analysis of navigation performance on the SHQ tasks and in real–world settings as a more objective and robust approach. The present study observed an attrition of five TBI participants and ten control participants, whereby the subsample of participants that completed the SHQ virtual navigation tasks was substantially less than that of the original sample that completed the SBSOD scale. It may be that the use of 're–contact surveys' is an appropriate strategy to provide additional insight into the missing data [38]. Such an approach would entail asking non–participants to complete the SHQ tasks. If such data is attainable, it may be analysed with reference to data of the subsample of participants that did complete the SHQ tasks to inform and potentially resolve power–loss resulting from the missing navigation performance data [39].

As is the case in the present study, the generalisability of conclusions drawn from samples of less than forty participants are inherently limited [40]. Therefore, future replication of this study in a larger sample with greater statistical power to detect a normal distribution and significant effect more reliably may strengthen the conclusions of the present study. A power analysis, with a large effect size of .08 indicates a sample of approximately 21 participants per group. Despite this, it should be noted that these limited sample sizes are often seen in TBI research of this nature (e.g., [2, 23, 24]). Furthermore, unlike in real–world navigation, sensory input used to guide navigation behaviour within a virtual simulation is limited to that of the visual system. Diersch and Wolbers explain that the perception of movement through space is only induced by optic flow during passive navigation and that the potential for transfer of spatial knowledge from body–based and visual–based cues may be limited by the differences in the encoding of VR versus real–world environments [19]. The naturalistic characteristics of the VR environment and the nature of the tasks are fundamental factors to consider when analysing a participant's navigation performance. The way in which these factors mediate one's interaction with the virtual environment is rarely discussed in the literature as a potential limitation to the validity or reliability of the measures. As such, subsequent work utilising immersive VR that enables both audio-visual but also vestibular and proprioceptive feedback on navigation may be warranted for future exploration.

## Conclusion

In conclusion, our study extends previous findings that have shown a spatial deficit in people with TBI. Patients spent a shorter duration viewing a map and took longer (in both distance and duration) completing the wayfinding levels. Path integration may be disrupted in this population if there is an absence of proximal cues. While a subjective measure (SBSOD) did not capture the navigation deficits within TBI patients, the Sea Hero Quest app did. Our exploratory research was conducted in collaboration with a charity and this has limited the numbers of participants and information and lesion locations. We report a preliminary analysis that we expect other research to follow up on to confirm. Our research therefore provides a promising avenue for exploring the effects of long-lasting clinical difficulties on navigation, using a mobile device.

## Supporting information

**S1 File. Raw data.**
(XLSX)

## Author Contributions

**Conceptualization:** Hugo J. Spiers, Rebecca Knight, Caroline Whyatt.

**Formal analysis:** Caroline Seton, Antoine Coutrot.

**Investigation:** Caroline Seton.

**Supervision:** Rebecca Knight, Caroline Whyatt.

**Visualization:** Antoine Coutrot.

**Writing – original draft:** Caroline Seton.

**Writing – review & editing:** Antoine Coutrot, Michael Hornberger, Hugo J. Spiers, Rebecca Knight, Caroline Whyatt.

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
