## [Decision Letter · Decision Letter 0]

22 Nov 2022

PONE-D-22-27835Wayfinding and path integration deficits detected using a virtual reality mobile app in patients with traumatic brain injuryPLOS ONE

Dear Dr. Knight, 

Thank you for submitting your manuscript to PLOS ONE. After careful consideration, we feel that it has merit but does not fully meet PLOS ONE’s publication criteria as it currently stands. Therefore, we invite you to submit a revised version of the manuscript that addresses the points raised during the review process.

Please submit your revised manuscript by Jan 06 2023 11:59PM . Please include the following items when submitting your revised manuscript:A rebuttal letter that responds to each point raised by the academic editor and reviewer(s). You should upload this letter as a separate file labeled 'Response to Reviewers'.A marked-up copy of your manuscript that highlights changes made to the original version. You should upload this as a separate file labeled 'Revised Manuscript with Track Changes'.An unmarked version of your revised paper without tracked changes. You should upload this as a separate file labeled 'Manuscript'.

We look forward to receiving your revised manuscript.

Kind regards,

Irene Cristofori

Academic Editor

PLOS ONE

2. Please ensure that you refer to Figure 6 in your text as, if accepted, production will need this reference to link the reader to the figure.

Reviewers' comments:

Reviewer's Responses to Questions

**Comments to the Author**

1. Is the manuscript technically sound, and do the data support the conclusions?

Reviewer #1: Partly

Reviewer #2: Yes

2. Has the statistical analysis been performed appropriately and rigorously? 

Reviewer #1: Yes

Reviewer #2: Yes

3. Have the authors made all data underlying the findings in their manuscript fully available?

Reviewer #1: Yes

Reviewer #2: Yes

4. Is the manuscript presented in an intelligible fashion and written in standard English?

Reviewer #1: Yes

Reviewer #2: Yes

5. Review Comments to the Author

Reviewer #1: Knight and colleagues investigated spatial-navigation abilities in a cohort of 15 patients with traumatic brain injury (TBI). Navigation abilities were self-estimated using the Santa Barbara Sense of Direction scale (SBSOD) and with the Sea Hero Quest VR test on mobile-app (SHQ). Only a sub-sample of subjects (10 TBI and 13 controls) participated to the SHQ test. All procedures were carried out remotely (online study). The SBSOD did not reveal any deficits in TBI (compared with 23 controls), while the SHQ test revealed that the TBI patients had impaired way-finding performance, as well as problems in path integration. The authors conclude that TBI can affect spatial-navigation abilities and that VR can be more sensitive compared with self-report measures.

The manuscript is well-written and easy to follow. Overall, the results are interesting and seem convincing (but see also the point below about sample size). Here are a few comments that may help strengthening the manuscript:

1. Extent/location of the brain injury. The manuscript does not include any objective information about the size/location of the lesion in the TBI patents. Rather, only self-reported location of the damage is included. This appears as a major weakness making it difficult to assess the level of (in)-homogeneity of the sample. I guess that the reason for the lack of these data is that the study was carried out remotely. I suggest that the Authors address disadvantages (and benefits) of this approach in the "limitations" section of the Discussion.

2. Sample size. The authors indicate that the sample size is "in line with previous research" (ln 185) and discuss this further in the "limitations" section of the Discussion. Nonetheless, the sample seems quite small and only a sub-sample of subjects participated to the SHQ task on which the main conclusions are drawn: did the authors carry out any power analysis?

3. Presentation of the design. The authors indicate that this was a 2x2 design with the factor groups (TBI/control) and task (wayfinding or path integration), ln 215. This seems somewhat misleading given that "wayfinding and path integration" apply only to the SHQ test (not to SBSOD) and that these two measures were not compared directly: i.e. they are not treated as levels of an experimental factor. Indeed the SHQ data were analysed using independent samples t-tests (ln 329). I suggest rephrasing the relevant section in the Methods.

4. No performance difference during training. The SHQ test included an initial training phase for which group comparisons did not reveal any difference (see also Fig 3). Why is that? If I understood correctly, also the training phase involved "wayfinding", so one may expect differences to arise also here. Is this a matter of the level-difficulty (but note that level-difficulty did not appear to systematically impact on the results of the main task, cf- ln 355)? Please discuss. Further analyses including "difficulty-level" as an experimental factor may also be added.

5. Effect of map-viewing. The authors suggest that TBI deficits in SHQ wayfinding may arise because of differences in map-viewing durations. However, Tab 3 shows that viewing times differed only for level 6 and 11, while Tab 2 indicated that wayfinding (duration) differed for levels 8 and 46. Please clarify.

Reviewer #2: This is a report on a non very frequently searched for deficit after TBI. The report is original, it is well written and deserves attention. I would rather describe it however as an exploratory study. It clearly shows with a small sample that TBI patients have long standing deficits in spatial orientation, and that specific active tests will detect them more than self reported questionnaires. I would suggest the authors not to determine in the methods section wether their sample is large enough to obtain significant results. Assesing the sample size should not be done using other studies for comparison. Another important point is the large number of comparisons made during the study and wether this was taking in to account. However the report is original enough to really be interesting and worth publishing.

6. PLOS authors have the option to publish the peer review history of their article (what does this mean?). If published, this will include your full peer review and any attached files.

Reviewer #1: No

Reviewer #2: **Yes: **Alfonso Lagares

---

## [Author Response · Author response to Decision Letter 0]

6 Jan 2023

1. Please ensure that your manuscript meets PLOS ONE's style requirements

File names for each figure have been changed. 

2. Please ensure that you refer to Figure 6 in your text as, if accepted, production will need this reference to link the reader to the figure.

Completed, line 453

3. Please include captions for your Supporting Information files at the end of your manuscript, and update any in-text citations to match accordingly. 

Completed, line 743

Reviewer Comments

Reviewer #1: 

1. Extent/location of the brain injury. The manuscript does not include any objective information about the size/location of the lesion in the TBI patients. Rather, only self-reported location of the damage is included. This appears as a major weakness making it difficult to assess the level of (in)-homogeneity of the sample. I guess that the reason for the lack of these data is that the study was carried out remotely. I suggest that the Authors address disadvantages (and benefits) of this approach in the "limitations'' section of the Discussion.

Response: The limitations of the self-reports are discussed in greater detail in lines 582 - 586. In the revisions we acknowledge the relative strengths and weaknesses of using this remote paradigm. 

“This limitation does make it difficult to assess the homogeneity of the sample. Despite the lack of objective measures, this study has allowed us to gain insight into spatial navigation impairment in TBI using a low cost, remote paradigm. In addition, the self-reported location of damage was not a significant predictor of the performance metrics in this study”. Line 582-586.

2. Sample size. The authors indicate that the sample size is "in line with previous research" (ln 185) and discuss this further in the "limitations" section of the Discussion. Nonetheless, the sample seems quite small and only a sub-sample of subjects participated to the SHQ task on which the main conclusions are drawn: did the authors carry out any power analysis?

Response: We have amended lines 183-187 to make it clearer that we are referring to the sub-sample numbers. 

“From the sample of thirty–eight participants that completed the SBSOD scale, a subsample of ten participants from the TBI group (n=10) and a subsample of thirteen participants from the control group (n=13) completed the SHQ navigation tasks. Thus, while the participant group remains small, this number is inline previous empirical studies; 14 TBI and 12 controls (23), TBI and 12 control (2), and eight TBI and 40 control (24)” Line 183-187

We have included a power calculation lines 606-608 and noted that our limited sample size constitutes a more “exploratory research” (line 147 & 627).

“A power analysis, with a large effect size of .08 indicates a sample of approximately 21 participants per group.” Line 606-608

3. Presentation of the design. The authors indicate that this was a 2x2 design with the factor groups (TBI/control) and task (wayfinding or path integration), ln 215. This seems somewhat misleading given that "wayfinding and path integration" apply only to the SHQ test (not to SBSOD) and that these two measures were not compared directly: i.e. they are not treated as levels of an experimental factor. Indeed the SHQ data were analysed using independent samples t-tests (ln 329). I suggest rephrasing the relevant section in the Methods.

Response: We have amended lines 220-232 to describe the comparison of the two groups (TBI and control) on three spatial navigation measures.

This study compared navigation ability between two groups (TBI or control) using three spatial navigation measures, namely the SHQ wayfinding duration and distance, SHQ path integration, and SBSOD scale. The first and second dependent variables, specific to performance on the wayfinding tasks, were duration (as measured by time taken in seconds to complete each level), and distance travelled (quantified as the Euclidean distance travelled between each sampled location in each level (in pixels)). Note, the coordinates of participants’ trajectories were sampled at Fs = 2 Hz. In line with the characterisation of wayfinding performance (12) better performance is represented by lower duration and distance metrics. The third dependent variable, specific to performance on the path integration tasks, was flare accuracy (i.e., selection of correct starting location in path integration tasks from three options). Lastly, the fourth dependent variable was participants’ scores on the fifteen–item SBSOD questionnaire, which range from 1 to 7, with higher scores indicating better self–rated navigational abilities (30). Line 220-232

4. No performance difference during training. The SHQ test included an initial training phase for which group comparisons did not reveal any difference (see also Fig 3). Why is that? If I understood correctly, also the training phase involved "wayfinding", so one may expect differences to arise also here. Is this a matter of the level-difficulty (but note that level-difficulty did not appear to systematically impact on the results of the main task, cf- ln 355)? Please discuss. Further analyses including "difficulty-level" as an experimental factor may also be added.

Response: The first two levels in SHQ have been designed to assess the participants’ ability to control the boat. These two levels are not wayfinding tasks. Performance on these two levels therefore give a “baseline” measure of how well the participant is able to use the interface. Clarification of this has been added to lines 272-274, 359-360.

“The first two levels completed by a participant were treated as training levels, as they were only designed to assess ability to control the boat (i.e., tap left to turn left, tap right to turn right, swipe up to speed up and swipe down to halt).” Line 272-274

“As these two training levels were not designed to assess wayfinding ability,” Line 359-360

5. Effect of map-viewing. The authors suggest that TBI deficits in SHQ wayfinding may arise because of differences in map-viewing durations. However, Tab 3 shows that viewing times differed only for level 6 and 11, while Tab 2 indicated that wayfinding (duration) differed for levels 8 and 46. Please clarify.

Response: Map view duration may be indicative of how accurately the map is encoded and therefore later recalled, as clarified in lines 396-397. 

“a factor which may relate to how accurately the map is encoded and later recalled” Line 396-397

There was a significant difference in overall map view duration, where the TBI group viewed for a shorter duration. We do however take the reviewer's comments regarding the level-by-level analysis on board, and have reflected this (lines 505-510). 

Although group differences in map view durations were significant overall, a level-by-level analysis indicates that the impact of these shorter encoding durations on wayfinding ability may be complex. Within this study, group differences in map viewing duration reached significance on levels 6 and 11 whereas group differences on wayfinding performance were significant for levels 8 and 46. Line 505-510

Reviewer #2: 

This is a report on a not very frequently searched for deficit after TBI. The report is original, it is well written and deserves attention. I would rather describe it however as an exploratory study. It clearly shows with a small sample that TBI patients have long standing deficits in spatial orientation, and that specific active tests will detect them more than self reported questionnaires. 

I would suggest the authors not to determine in the methods section whether their sample is large enough to obtain significant results. Assessing the sample size should not be done using other studies for comparison. Another important point is the large number of comparisons made during the study and whether this was taken into account. However the report is original enough to really be interesting and worth publishing.

Response: We have included a power calculation (lines 606-608) and noted that our limited sample size constitutes a more exploratory study (line 147-148 & 626-629).

“This present study was exploratory and conducted in collaboration with a charity in order to reach out to patients.” Lines 147-148

“Our exploratory research was conducted in collaboration with a charity and this has limited the numbers of participants and information and lesion locations. We report a preliminary analysis that we expect other research to follow up on to confirm.” Lines 626-629

---

## [Decision Letter · Decision Letter 1]

13 Feb 2023

Wayfinding and path integration deficits detected using a virtual reality mobile app in patients with traumatic brain injury

PONE-D-22-27835R1

Dear Dr. Knight,

We’re pleased to inform you that your manuscript has been judged scientifically suitable for publication and will be formally accepted for publication once it meets all outstanding technical requirements.

Kind regards,

Irene Cristofori

Academic Editor

PLOS ONE

Additional Editor Comments (optional):

Reviewers' comments:

Reviewer's Responses to Questions

**Comments to the Author**

1. If the authors have adequately addressed your comments raised in a previous round of review and you feel that this manuscript is now acceptable for publication, you may indicate that here to bypass the “Comments to the Author” section, enter your conflict of interest statement in the “Confidential to Editor” section, and submit your "Accept" recommendation.

Reviewer #1: All comments have been addressed

Reviewer #2: All comments have been addressed

2. Is the manuscript technically sound, and do the data support the conclusions?

Reviewer #1: Yes

Reviewer #2: Yes

3. Has the statistical analysis been performed appropriately and rigorously? 

Reviewer #1: Yes

Reviewer #2: Yes

4. Have the authors made all data underlying the findings in their manuscript fully available?

Reviewer #1: Yes

Reviewer #2: Yes

5. Is the manuscript presented in an intelligible fashion and written in standard English?

Reviewer #1: Yes

Reviewer #2: Yes

6. Review Comments to the Author

Reviewer #1: The authors have addressed my initial comments. In particular the manuscript now includes explicit statements concerning the exploratory/preliminary nature of the report.

Reviewer #2: Thanks for accepting the comments and make changes. Authors have addressed all comments. Tha manuscript is technically sound and is correctly presented.

7. PLOS authors have the option to publish the peer review history of their article (what does this mean?). If published, this will include your full peer review and any attached files.

Reviewer #1: No

Reviewer #2: **Yes: **Alfonso Lagares

---

## [Editor Report · Acceptance letter]

28 Feb 2023

PONE-D-22-27835R1 

Wayfinding and path integration deficits detected using a virtual reality mobile app in patients with traumatic brain injury 

Dear Dr. Knight:

I'm pleased to inform you that your manuscript has been deemed suitable for publication in PLOS ONE. Congratulations! Your manuscript is now with our production department. 

Kind regards, 

on behalf of

Dr. Irene Cristofori 

Academic Editor

PLOS ONE